# Food Habits and Lifestyle in Hyperphenylalaninemia Patients: Should These Be Monitored?

**DOI:** 10.3390/children9081164

**Published:** 2022-08-03

**Authors:** Annamaria Dicintio, Giulia Paterno, Rosa Carella, Federica Ortolani, Maristella Masciopinto, Donatella De Giovanni, Albina Tummolo

**Affiliations:** Department of Metabolic Diseases, Clinical Genetics and Diabetology, Giovanni XXIII Children Hospital, Azienda Ospedaliero-Universitaria Consorziale, 70126 Bari, Italy; annamaria.dicintio@gmail.com (A.D.); giupatvi@gmail.com (G.P.); rossycarella@gmail.com (R.C.); federicaortolani@hotmail.com (F.O.); m.masciopinto@libero.it (M.M.); degiovanni.dony@gmail.com (D.D.G.)

**Keywords:** HPA, body image, food neophobia, protein content

## Abstract

Studies on Hyperphenylalaninemia (HPA) patients are scarce and primarily focused on neurocognitive outcomes compared to PKU patients. In this study, we characterized the food habits and lifestyle of HPA patients compared with healthy peers. We performed a cross-sectional survey of a cohort of 30 patients (13 males, median age/range: 7.9; 2.2–16.7 years) and 28 controls (8 males, median age/range: 7.9; 2.1–16.7 years). Anthropometric parameters, food and nutrient intakes, and level of physical activity were assessed. Food neophobia, eating disorders, and body image perception was investigated by specific tests. Patients showed greater selectivity in the choice of foods than controls, preferring products with lower protein content (*p*-value: 0.03) and avoiding associating multiple protein and carbohydrate sources. A comparable tendency to distrust new foods emerged without elements suggestive of eating disorders. Patients had higher image dissatisfaction than peers (*p*-value: 0.01). This group of patients manifested more selective eating habits and worse body image acceptance. A regular evaluation of these aspects in these patients may result in a more effective follow-up of this disorder. More studies are needed to confirm these findings.

## 1. Introduction

Hyperphenylalaninemia (HPA) (OMIM 261600) is a recessive inherited metabolic condition caused by the inability to convert Phenylalanine (Phe) to tyrosine (Tyr) due to the total or partial defect of the enzyme phenylalanine hydroxylase (PAH) [1]. Classically, HPA has been distinguished in different clinical phenotypes based on the severity of the disease: classic Phenylketonuria (PKU) (with pre-treatment blood Phe levels over 1200 μmol/L), moderate PKU, and mild HPA as non-classic PKU (between 1200 and 360 umol/L) and untreated non-PKU HPA (between 120 and 360 umol/L) [2]. The most recent guidelines now prefer a simplified classification [3]. It divides patients into two groups: requiring or not requiring treatment.

The first group is made up of the most severe forms of the disease, requiring Phe-restriction and/or the use of pharmacological treatment. The latter is represented by mild HPA, a condition in which the Phe blood concentration remains within the therapeutic range without any protein restriction regimen. Due to the possibility of blood Phe concentration increasing with age, it has been reported that patients with Phe levels < 360 μmol/L should be monitored (at a lower frequency) during the first year of life as a minimum [4].

Reports on mild HPA patients are scarce and primarily focused on the neurocognitive outcome of untreated patients. There is general agreement that these patients’ neurological and psychological outcomes show no significant differences compared with healthy control subjects [5,6]. The anthropometry, body mass index (BMI), and clinical signs of nutrient intake of HPA patients, also appeared in the normal range [7].

Nevertheless, no data are available on the food habits and lifestyle of mild HPA patients on a normal diet.

Recent evidence focused on food behavioral alterations and low physical activity (PA) among PKU patients [8,9]. It has been observed that strict medical control over daily dietary habits can lead to attitudes towards eating disorders and negative body image, suggesting the importance of a tailored screening and awareness for eating disorders for PKU patients [9]. Among feeding disorders, food neophobia is defined as an attitude toward food, which manifests as a persistent reluctance to eat new foods [10] and plays a role in food refusal in PKU patients [11].

These issues have also been observed in patients affected by chronic diseases without strict dietetic limitations, such as diabetes type 1, in whom increased frequency of disorder eating behavior, such as fasting, voluntary dietary restriction, and body image disturbances, have been reported, as consequences of the clinical involvement of their nutrition [12,13]. In non-PKU HPA patients, the diagnosis of a genetic disease involving an altered protein metabolism, diagnosed at birth, could interfere with dietary behaviors and lifestyle.

In our study, we described for the first time food habits, level of physical activity, and body image perception of a cohort of non-PKU HPA children and adolescents following a normal diet. We compared them with a group of healthy controls.

## 2. Material and Methods

### 2.1. Subjects and Study Design

This is a cross-sectional study on children and adolescents affected by HPA not requiring therapy (in this study, referred to simply as HPA) and followed-up at a single center. The study was conducted between June 2020 and June 2021. Inclusion criteria were age between 2 and 17 years, diagnosis by newborn screening and confirmation by genetic analysis, regular follow-up, and free-diet regimen. Exclusion criteria were concomitant chronic conditions, gastroenteric disorders, and cognitive deficiencies. Thirty patients met the inclusion criteria and were enrolled in the study. They represented a significant sample size according to the incidence of HPA in our population (α error 0.05, statistical power 80%). A control group of 30 subjects with no significant differences in age and sex compared with patients was recruited among healthy volunteers. Two were excluded because they did not complete at least 75% of the tests, and 28 were included and analyzed.

Part of the study was conducted during the pandemic in Italy [14]. Therefore, in some cases, tests were performed via video consultation after sending via e-mail.

The study was approved by the Local Ethics committee. Informed consent was obtained from the parents of participants before the study.

### 2.2. Anthropometric Measurements

Patients were weighed and measured in minimal clothing by two dietitians using the same scale and stadiometer (Inbody 230-Wunder). Body mass index (BMI) was calculated as the weight/height^2^ ratio (kg/m^2^). Normal values of BMI Z-score and percentiles were calculated using the Cacciari charts [15]. Tanner stage (prepubertal, Tanner 1; pubertal, Tanner 2 to 5) was determined by physical examination [16].

### 2.3. Assessment of Food and Nutrient Intake

A food diary was administered to every patient for three days. Parents and patients were asked to describe breakfast, lunch, dinner, and eventual snacks, specifying place, time, duration, and people with them at that time. They reported particular situations at the moment of the meal (rejection of food, hunger or satiety at the end of the meal, selectivity or avoidance of food, vomiting, laments, worries, crying). On the basis of the food diaries, the dietitians determined the amount of daily caloric, nutrients, and Phe intakes using the Winfood Pro software (version 3.0.0, 2011, Medimatica SRL, Teramo, Italy). Daily protein, carbohydrate, and lipid intake were expressed in grams per day and grams per weight per day. Carbohydrate and lipid intake were also expressed as a percentage of total daily calories (% Kcal/total Kcal). They were compared with FAO/WHO/UNU 2007 requirements [17,18,19]. Daily phenylalanine intake was expressed in milligrams per day and milligrams per weight per day.

### 2.4. Food Neophobia and Eating Disorders Tests

The psychologist of the multidisciplinary team administered specific tests to investigate different aspects related to food behaviors, particularly the risk of food neophobia and eating disorders.

The risk of food neophobia was assessed using the Italian Child Food Neophobia Scale (ICFNS) [20], a self-report made up of 8 items to measure food neophobia in primary school children (6–9 years) (scores ≤ 17: “low neophobia”, scores ≥ 18 and ≤24: “medium neophobia”, scores ≥ 25: “high neophobia”). The Children Eating Attitude Test (Ch-EAT-26) [21] for children and preadolescents between 8 and 13 years and the Eating Attitude Test (EAT-26) [22] for adolescents > 13 years are self-report questionnaires made up of 26 items to measure the risk of eating disorder (abnormal scores ≥ 20).

### 2.5. Physical Activity

Physical activity was assessed using a question as a “proxy indicator of physical activity” through the question: “How would you describe your level of physical activity?” A 3-point Likert scale [23] was categorized as: (1) Almost no physical activity, (2) Moderate activity (walking or biking once/week), (3) Active physical activity (running, gym training 2–3 times/week).

### 2.6. Children Body Image Scale

Patients’ perception of their body image was investigated by the psychologist through the Children Body Image Scale (CBIS) [24], that consists in a representation of 7 male or female photos of children, every photo represents a value of BMI (1 = 3rd percentile; 2 = 10th percentile; 3 = 25th percentile; 4 = 50th percentile; 5 = 75th percentile; 6 = 85th percentile; 7 = >90th percentile in female photos and >97th percentile in male photos). This test was administrable to children from 7 to 12 years old. Chosen photos (perceived and desired body images) were compared with the actual BMI percentile of each subject. In our sample, 16 children and preadolescents in that age range were interviewed.

### 2.7. Statistical Analysis

Continuous variables were expressed as mean ± SD (standard deviation) or median and range and categorical variables as percentages (%). Two-sided Fisher’s exact and Mann–Whitney tests were used to compare categorical and continuous variables, respectively. The original *p*-value was adjusted for multiplicity by Benjamini and Hochberg method. *p*-Value < 0.05 was considered as statistically significant. Statistical analysis was performed using Microsoft^®^ Excel 2016 MSO, version 2206.

## 3. Results

### 3.1. Nutritional Parameters

Overall, the study sample comprised 30 patients, 13 males and 17 females (median age/range: 7.9/2.2–16.7 years) and 28 controls, 8 males and 20 females (median age/range: 7.9/2.1–16.7 years).

Weight and height were higher in patients than in controls, but BMI did not show significant differences in the SDS and percentile, falling within the normal range (+0.32 ± 1.18 vs. +0.02 ± 1.47) in both groups (Table 1).

Parents compiled diaries in all cases. The caloric intake was not significantly different in the two groups (Table 1), nor was the daily protein intake. The protein intake adjusted for weight and Phe intake per day was found to be lower in patients (*p*: 0.03 and *p*: 0.0004, respectively), who, instead tended to have a higher fat intake, also exceeding the WHO recommendations (33.2% Kcal/total Kcal vs. <30% of WHO recommendations) [17,18,19]. Carbohydrates were instead more consumed by controls than patients (*p*: 0.003), but both fell within the WHO % Kcal/total Kcal intake recommendations (45–60%) [17,18].

With regard to the composition of meals, foods consumed more than two times in three days are reported in Table 2. The main differences regarded lunch composition, with legumes as the first dish in controls and the second more often made up of dairy products or sausages for patients and meat or fish for the control group.

At dinner, they both preferred dairy products, but controls tended to eat more bread and other oven products than patients who preferred pasta or rice. Higher consumption of fizzy drinks and sweet food was noticed in controls.

### 3.2. Physical Activity

Report on PA (physical activity) was retrieved by a subtotal number of subjects (22 for the HPA group and 26 for the control group) because of missing data from some participants. The percentage of subjects who referred to “almost no physical activity” was comparable in the two groups (45% vs. 38%, *p*: 0.7) (Table 3). Among HPA subjects who declared regular PA, the majority performed “moderate physical activity” (50%) without a statistically significant difference with controls (*p*: 0.10). With regard to “active physical activity,” only 4% of patients declared to perform it, compared to 38% of the control group (*p*: 0.01).

### 3.3. Food Behavior and Body Image Assessment

Food behavior and body image data were obtained from all participants in the study.

The Italian Child Food Neophobia Scale, administered to patients (*n* = 10) and controls (*n* = 10) according to target age (6–9 years), demonstrated mean values in the range of “medium food neophobia” for both patients (21.9 ± 4.4) and control group (24.0 ± 7.7), without statistically significant difference between the two groups (*p*: 0.46).

Also, no risk of eating disorders, according to Ch-EAT/EAT tests administered to 15 patients and 14 controls, was detected in patients, with scores, under the upper limit in all the three age groups analyzed (6–9 years: 2.7 ± 1.5; 10–14 years: 10.6 ± 9; 14–17 years: 5 ± 6.2). In all cases, no significant differences with the control group were found (*p*: 0.17).

According to age, a body image test was administered to 16 patients (7 males and 9 females, median age: 9.91 and 14 controls (3 males and 11 females, median age: 9.90. The patient group showed a discrepancy between actual and perceived BMI percentile (*p*: 0.04) the latter resulting lower than the actual one, in almost all cases. An even higher discrepancy was found between desired and perceived image (*p*: 0.01) and between the actual and desired body image (*p*: 0.0001) (Figure 1).

The Control group manifested a better awareness of body image, with higher similarity between actual and perceived BMI (*p*: 0.31). The desired BMI was lower than the perceived one, although closer than in the HPA group (*p*: 0.06). Also, this group manifested a discrepancy between desired and real BMI (*p*: 0.03). In both groups, a better awareness of self-image was noticed by lower BMI, and no differences between males and females were detected.

## 4. Discussion

This study explores for the first time the eating habits, physical activity, and the self-perception of HPA patients on a normal diet. This group of patients appears to limit the use of protein-rich foods, avoiding associating multiple protein and carbohydrate sources in the same meal. An equal tendency between patients and controls to distrust new foods was noticed, without elements suggestive of eating disorders. Patients tended to perform a lower level of physical activity, almost avoiding intense physical activity, and manifested higher body image dissatisfaction than peers.

Disordered eating behavior has already been observed in chronic conditions such as cystic fibrosis, celiac disease, and inflammatory bowel diseases, in which the necessity to remain on a special diet may trigger disordered eating behaviors or eating selectivity [25,26]. In PKU patients, neophobia appears to play a significant part in food refusal by patients [11]. The lower protein intake may alter body composition, negatively impacting body image [27].

However, many recent studies have emphasized a high incidence of food selectivity and neophobia among children and adolescents without a chronic condition; the health consequences of food neophobia consist mainly in the potential reduction of the consumption of products containing necessary nutrients due to an imbalanced diet [10]. In this regard, great emphasis has been reserved for planned diets considered healthier and preferable to a free-diet regimen, even without a confirmed food intolerance [28].

Results from this study, highlighting a medium level of food neophobia in both patients and healthy subjects, are consistent with the concern of food selectivity and neophobia that are highly impacting our society [29], especially among children [30] and adolescent subjects [31]. The scales used in this study are validated for other conditions [32] as no specific tests are so far available for HPA patients. Indeed, various studies have pointed out the heterogeneity of tests used to assess food neophobia in children and adolescents [29,33], raising the need for a higher uniformity of tools to investigate these aspects, both for the general population and for subjects affected by chronic conditions.

In this patients group, the diagnosis of a genetic disease involving an altered protein metabolism, diagnosed at birth, could have led to a more or less conscious restriction of protein intake, particularly of Phe intake, as demonstrated by a sort of avoidance of Phe rich food as for the dietary diaries (legumes, fish, meat). Also, a parental influence on the eating habits of these subjects may have played a role, as already highlighted by the literature [33].

No specific guidelines are so far available on the type and frequency of follow-up in HPA patients. In our center, we performed a regular follow-up for the first year of age, assessing growth and type of nutrition until a stable Phe tolerance is reached. Based on the Phe values monitoring, a free-regimen diet is promoted in this period. Thereafter, a year-based follow-up is settled, generally until adolescence, with higher frequency only in case of particular comorbidities or emerging problems.

The incidence and the role of physical activity on metabolic control and long-term outcome in both PKU and HPA patients have been poorly investigated. A study by Jani et al. [8] highlighted that PKU patients mostly perform moderate physical activity, and only a lower percentage (15% of adults and 45.9% of children) perform intense physical activity, which was associated with a reduction of body fat mass.

Our study found a very low rate of active physical activity among patients, whereas controls resulted in more activity, despite the same lockdown condition. Maybe differences in physical activity between PKU or HPA patients and healthy subjects could relate to the genetic diagnosis itself that discourages parents from dealing with their children as the others, underestimating their possibilities. Over time, children and adolescents become more vulnerable in perceiving their body image and physical competency. Recently, systematic reviews have pointed out that body image in adolescence becomes a significant determinant of physical activity attrition more than actual physical skills [34,35].

Body image refers to a complex psychological experience and includes thoughts and beliefs about one own appearance (memories, assumptions, generalizations) and body, including height, shape, and weight.

Young people with chronic illness have a less positive body image than their healthy peers [36]. Chronic illness, such as diabetes, impacts the body and the psyche, disturbing the body image of the child or adolescent. This must be considered by professionals who deal with their growing up issues [37].

Many studies analyzed body size estimation in the young population, associated with age, gender, and BMI, highlighting the relationship between underestimation and overweight [38,39,40]. Healthy children and adolescents with higher BMI are more likely to underestimate their body size, whereas low weight predicts overestimation [39,40]. Many internalize messages, starting at a young age, can lead to adverse psychological effects later in life, including anxiety, depression, and low self-esteem; here because psychological, sociocultural, and maturational factors must be generally considered in the young population’s well-being [41].

This study has some limitations. It was partially conducted over the pandemic, impacting food habits and physical activity. This limitation was partly overcome by the fact that patients and controls were examined in the same contextual conditions. Furthermore, self-reported bias cannot be ruled out because the diet behaviors and physical activity data were self-reported, introducing possible bias. Finally, the sample is small, and the number of patients answering some of the tools differs, leading to a comparison of different samples in some cases.

This study suggests that patients with HPA may be more prone to have selective eating habits and worse acceptance of their body image than peers. The constant promotion of a free-diet regimen associated with monitoring behaviors and thoughts could increase patient awareness and improve disease management.

Further studies on these aspects and comparing HPA with PKU patients must confirm these findings.

## Figures and Tables

**Figure 1 children-09-01164-f001:**
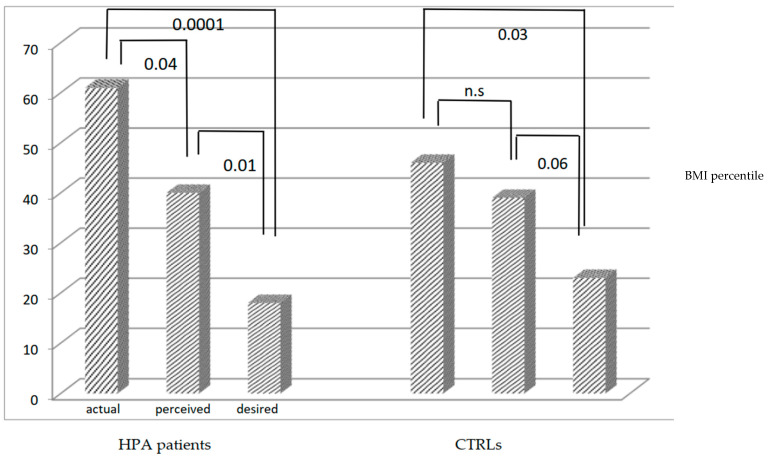
Differences among actual/perceived/desired BMI in HPA patients and controls. n.s. = not significant. HPA: Hyperphenylalaninemia; CTRLs: controls.

**Table 1 children-09-01164-t001:** Characteristics of HPA patients and control groups.

Demographics	Non-PKU HPA Group	Control Group	*p*-Value
	*n* = 30	*n* = 28	
**Median age (range)**	7.9 (2.2–16.7)	7.9 (2.1–16.7)	n.s.
**Number of subjects by age range**			
Range 2–5	9	9	n.s.
Range 6–9	10	10	n.s.
Range 10–13	8	6	n.s.
Range 14–17	3	3	n.s.
Gender (M/F)	13/17	8/20	n.s.
**Gender by age group (M/F):**			
Range 2–5	4/5	4/5	n.s.
Range 6–9	3/7	1/9	n.s.
Range 10–13	4/4	2/4	n.s.
Range 14–17	2/1	1/2	n.s.
Ethnicity:			
Caucasian	30/30	28/28	n.s.
**Anthropometry (mean ± SD)**			
Heights SDS	+0.07 ± 1.06	−0.51 ± 1.35	0.03
Weight SDS	+0.24 ± 1.29	−0.27 ± 1.47	n.s.
BMI (kg/m^2^)	19.64 ± 5.192	18.697 ± 4.72	n.s.
BMI SDS	+0.32 ± 1.18	+0.02 ± 1.47	n.s.
BMI percentile	56.23 ± 33.16	52.576 ± 34.81	n.s.
**Nutritional intake (mean ± SD)**			
Caloric intake (kcal/day)	1293.01 ± 334.12	1428.91 ± 376.75	n.s.
Caloric intake (kcal/Kg/day)	47.50 ± 27	57 ± 26.36	n.s.
Protein intake (gr/day)	52.05 ± 15.60	52.11 ± 13.63	n.s.
Protein intake (gr/kg/day)	1.93 ± 1.15	2.14 ± 1.12	0.03
Phenylalanine intake (mg/day)	1732.06 ± 581.50	2347.40 ± 603.40	0.0004
Phenylalanine intake (mg/kg/day)	68.86 ± 42.89	97.47 ± 51.98	0.02
Carbohydrate intake (gr/day)	166.182 ± 52.33	212.394 ± 61.071	0.003
Carbohydrate intake (% Kcal/total Kcal)	51.495 ± 8.65	59.273 ± 5.71	0.0002
Fat intake (gr/day)	47.95 ± 16.394	46.68 ± 14.52	n.s.
Fat intake (% Kcal/total Kcal)	33.152 ± 5.996	27.10 ± 5.288.5	0.003

n.s. = not significant.

**Table 2 children-09-01164-t002:** Most consumed food analyzed by three-day food diaries in patients and controls.

	Hyperphenylalaninemia Patients	Controls
	**Most Consumed Foods** *	**Most Consumed Foods** *
**Breakfast**	milk or yogurt, cereals, and derivates	milk or yogurt, cereals, and derivates
**Mid-morning snack**	fruit juice or salty snacks	sweets
**Lunch**		
**first course**	pasta or rice with vegetables	pasta or rice with vegetables/legumes
**second course**	dairy products or sausages	fish, meat
**dessert**	fruit, sweets	fruit, sweets
**Mid-afternoon snack**	Sweets	sweets
**Dinner**		
**first course**	pasta or rice	
**second course**	vegetables, dairy products, or cheeses	vegetables, dairy products or cheeses, bread, and other oven products
	fruit	fruit, sweets
**dessert**		
**Bed-time snack**	not routinely consumed	not routinely consumed
**Drinks**	water, fruit juice	water, fizzy drink

* ≥2 times in 3 days.

**Table 3 children-09-01164-t003:** Level of Physical activity (PA) in patients and controls.

Level of PA (Number/%)	HPA (22)	CTRLs (26)	*p*-Value *
Almost no PA	10 (45)	10 (38)	0.62
Moderate PA	11 (50)	6 (23)	0.10
Active PA	1 (4.5)	10 (38)	0.01

* adjusted for multiplicity by Benjamini and Hochberg method. PA: physical activity; CTRLs: controls.

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
