# Peer review of "Food Habits and Lifestyle in Hyperphenylalaninemia Patients: Should These Be Monitored?"

_children, 2022, doi:10.3390/children9081164_

Round 1

Reviewer 1 Report

General comments

The manuscript titled “Food habits and lifestyle in non-PKU Hyperphenylalaninemia patients: should these be monitored?” evaluates an interesting topic and provides original results.  The study highlighted higher image dissatisfaction in non-PKU hyperphenylalaninemia cases versus peers. Nevertheless, various aspects should be improved, as detailed in the specific comments below.

Specific Comments:

Abstract

The relevant quantitative data and p-values should be added to substantiate results. The allegation about greater food selectivity in patients is not clearly linked to the results (Table 2).

Introduction: OK

Section 2.1. Subjects and study design.

-A major limitation of this study pertains to the small sample size (30 cases and 28 controls). The authors should present a power calculation for the main outcome. What was the power of the study?

-The authors state that  “Altogether of the 75 patients, enrolled in the study, 30 met the inclusion criteria and were included.” The authors should provide details about the specific reasons that resulted in 45 patients excluded from the study.

-Most frequently, cases and controls are 1:1 matched. Why did the authors include 28 instead of 30 controls? Were there any refusals of participation?

-Age matching most frequently implies +/- some months; the authors should further elaborate on age matching in their study.

-There seems to be a fundamental issue regarding sex matching. The authors stated that they performed also matching on sex; however, they included 13 male and 17 female cases versus 8 male and 20 female controls (cf. also section 3.1). There seems to be no sex matching; therefore the relevant statement should be eliminated.

-Protocol number registration should be provided regarding the approval by the Local Ethics Committee

Section 2.3. Assessment of Food and Nutrient Intake

-The authors stated that parents and patients provided information about nutritional habits. For how many subjects was information based on parental reporting? For how many on self-reporting? Were there subjects for which parental and self-reporting were present? What was the degree of agreement between them? (kappa statistic)

Section 2.5. Physical activity

-The definition of moderate and active physical activity is vague. The authors should provide details about how they defined “moderate” and “active physical activity”.

-Why did the authors provide data on physical activity only on 22 cases and 26 controls? The authors should explain this discrepancy, as it might denote selection bias.

2.7. Statistical analysis

Please correct: “categorical and continuous variables” instead of “categorical and continuous parameters”

3.1. Nutritional and physical activity parameters

-The title should be amended, as no details on physical activity were provided in this section.

-“CHO” should be spelled out in Table 1.

-Lining has been distorted in “gender by age group” and “ethnicity” in Table 1

-“A higher consume” (line 151) should be replaced by “A higher consumption”

-The asterisk in Table 2 appears only in the HPA group; does it also apply in the CTRL group? If yes, it should be added in the respective column title.

-As mentioned above (Abstract), Table 2 does not clearly document food selectivity. Was there a statistical test to substantiate differences between the two groups? 

3.2. Physical activity

-In Table 3, the authors provide separate p-values for each physical activity level. This implies some form of post hoc merging of categories and the level of statistical significance should be adjusted due to multiple comparisons.  Please consult the following methodological reference and amend the level of statistical significance:

Austin PC, Goldwasser MA. Pisces did not have increased heart failure: data-driven comparisons of binary proportions between levels of a categorical variable can result in incorrect statistical significance levels. J Clin Epidemiol. 2008 Mar;61(3):295-300. doi: 10.1016/j.jclinepi.2007.05.007.

3.3. Food behavior and body image assessment

Paragraph 3: The descriptive statistics of actual, perceived and desired BMI percentiles should be added, accompanying the relevant p-values.

4. Discussion

Interesting and well written; it would benefit from a commentary regarding the limited statistical power of the study stemming from the small sample size.

Author Response

Section 2.1. Subjects and study design.

-A major limitation of this study pertains to the small sample size (30 cases and 28 controls). The authors should present a power calculation for the main outcome. What was the power of the study?

The sample size was calculated in the basis of the incidence of HPA among our patients with a power of 80%. We calculated that 30 subjects still represent a significant sample size. It is detailed in the text.

-The authors state that “Altogether of the 75 patients, enrolled in the study, 30 met the inclusion criteria and were included.” The authors should provide details about the specific reasons that resulted in 45 patients excluded from the study.

They did not met inclusion criteria, anyway we clarified the sentence in the text (lines 75-77)

-Most frequently, cases and controls are 1:1 matched. Why did the authors include 28 instead of 30 controls? Were there any refusals of participation?

Reason for the 28 subjects instead of 30, is now detailed in the text

-Age matching most frequently implies +/- some months; the authors should further elaborate on age matching in their study.

We confirm age matching between the two groups, we considered median and ranges and not mean +/- SD, due to the distribution of the parameter.

-There seems to be a fundamental issue regarding sex matching. The authors stated that they performed also matching on sex; however, they included 13 male and 17 female cases versus 8 male and 20 female controls (cf. also section 3.1). There seems to be no sex matching; therefore the relevant statement should be eliminated.

We now rewrited the sentence, referring to the lack of statistical significant difference between the two parameters (lines 78-79).

-Protocol number registration should be provided regarding the approval by the Local Ethics Committee

The protocol number assigned to the study by the Policlinico of Bari Ethic Committee is: 66418

Section 2.3. Assessment of Food and Nutrient Intake

-The authors stated that parents and patients provided information about nutritional habits. For how many subjects was information based on parental reporting? For how many on self-reporting? Were there subjects for which parental and self-reporting were present? What was the degree of agreement between them? (kappa statistic)

Diaries were completed in all cases by parents, we specified it in the manuscript.

Section 2.5. Physical activity

-The definition of moderate and active physical activity is vague. The authors should provide details about how they defined “moderate” and “active physical activity”.

Thanks for this question, we have added details on that at lines 121-123.

-Why did the authors provide data on physical activity only on 22 cases and 26 controls? The authors should explain this discrepancy, as it might denote selection bias.

 The discrepancy was now explained at lines 172-173.

2.7. Statistical analysis

Please correct: “categorical and continuous variables” instead of “categorical and continuous parameters”

Done.

3.1. Nutritional and physical activity parameters

-The title should be amended, as no details on physical activity were provided in this section.

Done.

-“CHO” should be spelled out in Table 1.

Done.

-Lining has been distorted in “gender by age group” and “ethnicity” in Table 1

Done.

-“A higher consume” (line 151) should be replaced by “A higher consumption”

Done

-The asterisk in Table 2 appears only in the HPA group; does it also apply in the CTRL group? If yes, it should be added in the respective column title.

Yes, it has been added.

-As mentioned above (Abstract), Table 2 does not clearly document food selectivity. Was there a statistical test to substantiate differences between the two groups? 

Table 2 highlights the main differences in composition of meals between the two groups, therefore it represents a qualitative comparison of different choices, for example: less legumes, fish and meat in the patient group, more sweets in the control group. In this context, no statistical tests could be used.

3.2. Physical activity

-In Table 3, the authors provide separate p-values for each physical activity level. This implies some form of post hoc merging of categories and the level of statistical significance should be adjusted due to multiple comparisons. Please consult the following methodological reference and amend the level of statistical significance:

Austin PC, Goldwasser MA. Pisces did not have increased heart failure: data-driven comparisons of binary proportions between levels of a categorical variable can result in incorrect statistical significance levels. J Clin Epidemiol. 2008 Mar;61(3):295-300. doi: 10.1016/j.jclinepi.2007.05.007.

We understand that data analysis was not complete, we are sorry for that. Now we inserted the corrected p-values for each categorical parameter. The original p-value was adjusted for multiplicity by Benjamini and Hochberg method. Slight different p-values were obtained (see table 3).

3.3. Food behavior and body image assessment

Paragraph 3: The descriptive statistics of actual, perceived and desired BMI percentiles should be added, accompanying the relevant p-values.

We added the descriptive statistics for the 16 patients and 14 controls (see lines 194-196). We can not add descriptive statistics for the three groups of BMI (actual, perceived, desired), as these parameters were provided by each participant to the study.

  1. Discussion

Interesting and well written; it would benefit from a commentary regarding the limited statistical power of the study stemming from the small sample size.

We have included this aspect among the limitations of the study.

Reviewer 2 Report

Thank you for your paper on a very interesting, and little researched topic. I enjoyed reading it. My only comment is that i would like a paragraph on how the clinical metabolic team manages nonPKU Hyperphe. How often do they see them? What information do you give them? Do you measure phe levels on a regular basis? I suspect the potential negative consequences are very dependent on what they are told and how they are managed. Also perhaps a little more re what we can do to prevent these negative outcomes   

Author Response

Thank you for your paper on a very interesting, and little researched topic. I enjoyed reading it. My only comment is that i would like a paragraph on how the clinical metabolic team manages nonPKU Hyperphe. How often do they see them? What information do you give them? Do you measure phe levels on a regular basis? I suspect the potential negative consequences are very dependent on what they are told and how they are managed. Also perhaps a little more re what we can do to prevent these negative outcomes   

Dear reviewer, thank you for this comment, this is an interesting point. We expanded on that in the discussion section at lines 245-250, 290-294.

Reviewer 3 Report

This is an interesting piece of work, and I would like to congratulate the authors for their effort putting this data together. I believe this manuscript needs some more work. I would suggest that some rewriting is needed as some sentences are not clear and some more ideas should be added to the discussion. I will give some specific suggestions below:

-      Non-PKU hyperphenylalaninemia: I think this is confusing. Usually, patients that are diagnosed with blood Phe levels (120-360 µmol/L) are usually described in the literature by HPA (hyperphenylalaninemia). I would suggest changing it throughout the manuscript.

Introduction

Line 47 - should read “… no data are available on food habits and lifestyle of non-PKU HPA patients on normal diet.”

Line 49 - should read “Recent evidence focused…”

Line 50 - should read “…PKU patients”

Line 58 – should read “…disorder eating behaviour”…

Line 65 - should read “…following a normal diet.”…

Should describe treatment of HPA in your clinic compared with PKU

Methods

Comment line 68-74: need to specify that control group are healthy volunteers and how they were recruited…

Line 73: should be something like: “Out of 75 patients screened, 30 met inclusion criteria and were enrolled in the study.”

Line 75: should read – “…pandemic in Italy… after sending via e-mail.”

Comment line 68-74: I think you need to add reference number of the approval given by ethics committee

Line 82: “… he same scale and stadiometer…”

I also think you need to add patient numbers for each tool used as they are all different. I also don’t understand that if this is a cross-sectional study how do you have such a different number of questionnaires done with each tool?

Table 1: - Is BMI percentile really needed?

-      Gr should be g

-      You should use the same number of decimal cases throughout the table

-      Please add abbreviations at the end of the table

Line 143: Maybe add comparison with protein recommendations…

Table 2. Most consumed foods analyse by three-day food diaries in patients and controls.

-      Table 2. Either add abbreviations at the bottom or just right full words…

Line 167: same number of decimal cases.

Line 188 – should read “… HPA patients on a normal diet.”

Line 201 to 204: Please rewrite as the idea is not totally clear. Also, I think this could be explored a bit more as there are no difference in dietary intake…

Line 220-224: I think could discuss a bit more if diet is not different why the differences in physical activity?

Line 229 – 235: I think could discuss a bit more body image compared with other chronic illnesses and general population?

Add to limitations that the number of patients answering to the different tools is different, so we are always comparing different samples.

General comments: It could be interesting to study body image in classic PKU patients as in theory they will be much different than the HPA compared with the normal population

Maybe give suggestions of what should be done to address you results in terms of monitoring and perhaps how to solve the issues

Author Response

This is an interesting piece of work, and I would like to congratulate the authors for their effort putting this data together. I believe this manuscript needs some more work. I would suggest that some rewriting is needed as some sentences are not clear and some more ideas should be added to the discussion. I will give some specific suggestions below:

Non-PKU hyperphenylalaninemia: I think this is confusing. Usually, patients that are diagnosed with blood Phe levels (120-360 µmol/L) are usually described in the literature by HPA (hyperphenylalaninemia). I would suggest changing it throughout the manuscript.

Change has been done, specifying the type of HPA we refer to.

Introduction

Line 47 - should read “… no data are available on food habits and lifestyle of non-PKU HPA patients on normal diet.”

Done

Line 49 - should read “Recent evidence focused…”

Done

Line 50 - should read “…PKU patients”

Done

Line 58 – should read “…disorder eating behaviour”…

Done

Line 65 - should read “…following a normal diet.”…

Done

Should describe treatment of HPA in your clinic compared with PKU

Added in the discussion, lines 245-250.

Methods

Comment line 68-74: need to specify that control group are healthy volunteers and how they were recruited…

We specified that in lines 78-81.

Line 73: should be something like: “Out of 75 patients screened, 30 met inclusion criteria and were enrolled in the study.”

We corrected this sentence in a clearer one.

Line 75: should read – “…pandemic in Italy… after sending via e-mail.”

Done

Comment line 68-74: I think you need to add reference number of the approval given by ethics committee

We have got the protocol number assigned to the study by the Policlinico of Bari Ethic Committee, that is: 66418. Official approval will be obtained in a few weeks.

Line 82: “… he same scale and stadiometer…”

Done

I also think you need to add patient numbers for each tool used as they are all different. I also don’t understand that if this is a cross-sectional study how do you have such a different number of questionnaires done with each tool?

At lines 184-185 we specified that the undrelied parameters were obtained by all partecipants. We also added the number of subjects for Ch-EAT/EAT tests (lines 190-191).

Table 1: - Is BMI percentile really needed?

We used it for comparisons with CBIS test, it’s necessary for our results

-      Gr should be g

Done

-      You should use the same number of decimal cases throughout the table

Done

-      Please add abbreviations at the end of the table

Done

Line 143: Maybe add comparison with protein recommendations…

We added a reference to make a comparison.

Table 2. Most consumed foods analyse by three-day food diaries in patients and controls.

Done

-      Table 2. Either add abbreviations at the bottom or just right full words…

We changed in full words.

Line 167: same number of decimal cases.

Done

Line 188 – should read “… HPA patients on a normal diet.”

Done

Line 195-197: Please rewrite as the idea is not totally clear. Also, I think this could be explored a bit more as there are no difference in dietary intake…

We rewrited the sentence in a clearer way, however Phe, protein intake/kg and carbohydrate intake are actual significantly different between the two groups.

Line 220-224: I think could discuss a bit more if diet is not different why the differences in physical activity?

We have added more info on that, see lines 257-264.

Line 229 – 235: I think could discuss a bit more body image compared with other chronic illnesses and general population?

Done at lines 268-280.

Add to limitations that the number of patients answering to the different tools is different, so we are always comparing different samples.

Done

General comments: It could be interesting to study body image in classic PKU patients as in theory they will be much different than the HPA compared with the normal population

Maybe give suggestions of what should be done to address you results in terms of monitoring and perhaps how to solve the issues

We addressed these issues at lines 290-299.

Round 2

Reviewer 1 Report

The authors satisfactorily revised their manuscript.